



# Regulation of N₂O emissions from acid organic soil drained for agriculture: Effects of land use and season

**Taghizadeh-Toosi, Arezoo[1], Elsgaard, Lars[1], Clough, Tim[2], Labouriau, Rodrigo[3] & Petersen, Søren Ole[1]**

[1] Department of Agroecology, Aarhus University, Tjele, 8830, Denmark

[2] Faculty of Agriculture and Life Sciences, Lincoln University, Christchurch, New Zealand

[3] Applied Statistics Laboratory, Department of Mathematics, Aarhus University, Aarhus, Denmark

*Correspondence to*: Arezoo Taghizadeh-Toosi (Arezoo.Taghizadeh-Toosi@agro.au.dk)

**Abstract**

Organic soils are extensively under agricultural management for cereal and high-value cash crop production or as grazing land. Drainage and tillage is known to promote emissions of nitrous oxide (N₂O), however, a previous monitoring program found that, in addition to effects of land use, annual N₂O emissions from fields with rotational grass and potato showed distinct seasonal patterns. A new study was therefore conducted to investigate the regulation of N₂O emissions in an area with raised bog and which was previously classified as a potentially acid sulfate soil. Four sites, i.e., two sites with rotational grass and two with a potato crop, were equipped for weekly monitoring of soil surface N₂O emissions and sub-soil N₂O concentrations to 1 m depth during spring and autumn 2015. Also, various environmental variables (precipitation, air and soil temperature, soil moisture, water table (WT) depth, and soil mineral N) were recorded. In late April and early September 2015, intact cores to 1 m depth were further collected at adjacent grassland and potato sites and analysed for pH, EC, nitrite ($NO_2^-$), total reactive Fe (TRFe), acid volatile S (AVS) and chromium-reducible S (CRS). The soil pH varied between 4.6 and 5.5. Total N₂O emissions during 152-174 days were 4-10 kg N₂O ha$^{-1}$ for rotational grass, and 30-32 kg N₂O ha$^{-1}$ for arable sites with a potato crop. Soil N₂O concentrations ranged from around 10 µL L$^{-1}$ at grassland sites to several hundred µL L$^{-1}$ at 50-100 cm depth at sites with potato. This reflected lower soil mineral N concentrations at grassland sites where probably competition from plants for available N was effective. Fertilisation had no immediate effect on N₂O emissions, but effects appeared in connection with rainfall where the WT also rose toward the soil surface and N₂O accumulated in the soil profile at all sites. Graphical models showed that the strongest predictor for N₂O emissions from both grassland and potato sites in spring, and grassland sites in autumn, was soil N₂O concentration near the WT depth. In contrast, for potato sites in autumn, nitrate ($NO_3^-$) in the top soil, together with temperature, controlled N₂O emissions. The distribution of TRFe and $NO_2^-$ in soil profiles suggested that chemodenitrification in the capillary fringe could be a significant source of N₂O during WT drawdown in spring, while N₂O emissions associated with the rapid soil wetting and WT rise in autumn may be attributed to biological denitrification. The concentration of TRFe in soil profiles was related to soil organic carbon, and much higher than concentrations of AVS, and thus iron oxides/hydroxides rather than iron sulfides were probably the source





of TRFe. Controlling seasonal WT dynamics and soil mineral N accumulation appear to be important controls of $N_2O$ emissions in acid organic soil used for agriculture.

**Key words:** Organic soil, potentially acid sulfate soil, rotational grass, potato, nitrous oxide, reactive iron

## 1 Introduction

According to the definition of the Food and Agriculture Organisation of the United Nations, organic soils (Histosols) must contain ≥ 12 percent or, if drained, > 20 percent organic carbon (C) at 0-20 cm depth. Over 50 percent of the soil

organic C stocks in Europe are stored in organic soils (peatlands) where historically, due to water-saturated and predominantly anaerobic conditions, organic matter has accumulated (Schils et al., 2008). Agricultural use of organic soil requires drainage, and this accelerates decomposition of soil organic matter and net C and nitrogen (N) mineralisation above the water table (WT) depth (Schothorst, 1977). It makes drained organic soils a significant net source of greenhouse gas emissions (GHG), mainly due to carbon dioxide ($CO_2$) and nitrous oxide ($N_2O$) emissions

(Goldberg et al., 2010; Maljanen et al., 2003).

Worldwide, 25.5 million ha with organic soil has been drained for agricultural use, mainly as cropland, according to Tubiello et al. (2016). Recently a supplement to the 2006 IPCC Guidelines for National Greenhouse Gas Inventories on Wetlands (IPCC, 2013) proposed average annual emission factors of 4.3 and 8.2 kg $N_2O$-N ha$^{-1}$ yr$^{-1}$ for grassland on drained organic soil of low and high nutrient status, respectively, and an emission factor of 13 kg $N_2O$-N ha$^{-1}$ yr$^{-1}$ for

cropland. For soil C losses the effect of land use is smaller, with emission factors for the three land use categories ranging between 5.3 and 7.9 Mg $CO_2$-C ha$^{-1}$ yr$^{-1}$ (IPCC, 2013). This indicates that site conditions are more critical for $N_2O$ than for $CO_2$ emissions. Site conditions are defined by land use, management, inherent soil properties and climate (Mander et al., 2010; Leppelt et al., 2014). Maljanen et al. (2003) found that WT depth, $CO_2$ emissions and temperature at 5 cm depth together explained 55% of the observed variability in $N_2O$ emissions during a two-year field study on a

drained organic soil, whereas the response to N fertilisation was limited, and they suggested that N released from soil organic matter was the main source of $N_2O$. Petersen et al. (2012) also found, in a study comparing GHG emissions from different land uses in three regions that site conditions such as groundwater level, pH and precipitation contributed significantly to explain $N_2O$ emissions.

The seasonal dynamics of environmental conditions such as temperature, precipitation and WT depth may be

considerable, and investigating relationships between potential driving variables and $N_2O$ emissions in transition periods could thus help identify sources. In acid organic soil, a number of pathways can lead to $N_2O$ formation that are associated with biotic or abiotic nitrification or denitrification under aerobic or anaerobic conditions (Braker and Conrad, 2011; Herrmann et al., 2012; Jones et al., 2015; Maeda et al., 2015; Spott et al., 2011). Characterising soil profiles with respect to potential electron donors and acceptors of putative chemical or microbial processes can further

inform about the importance of potential sources of $N_2O$.





We studied four agricultural sites within a raised bog area with acid soil conditions, two sites with a potato crop in the experimental year and two sites with rotational grass. The study covered spring and autumn periods where high emissions were previously observed (Petersen et al., 2012). We searched for relationships between seasonal variation in $N_2O$ emissions and potential driving variables in order to isolate effects and interactions of such factors as temperature,

precipitation, WT depth and N availability. We pursued the hypotheses that $N_2O$ would be produced in the capillary fringe in periods with fluctuating WT, possibly with links to inorganic redox transformations, and that emissions of $N_2O$ would be higher from the arable crop than from grassland.

## 2 Materials and methods

### 2.1 Study sites

The sites selected for this study were all located in Store Vildmose which is a 5,000 ha raised bog in northern Denmark. The area was, until 150 years ago, the largest raised bog in Denmark, and largely unaffected by human activity. The bog overlies a marine plain formed by the last marine transgression; as the sea retreated around 8000 BC, peat developed in wet parts of the landscape attaining a maximum depth of 4.5 to 5.3 m in central parts of the bog (Kristensen, 1945). Between 1880 and 2010, the peat has generally subsided by at least 2 m due to drainage for agriculture or peat

excavation (Regina et al., 2015), and today the peat depth is generally 1-2 m or even less. The peat and underlying sand is acidic and has been characterised as a potentially acid sulfate soil due to a recorded potential for oxidation of pyrite when drained (Madsen and Jensen, 1988).

Four field sites were distributed along an east-west transect after field trips and meetings with farmers. Two of these sites, located side by side, were also represented in a monitoring program conducted in 2008-2009 (Petersen et al.,

2012). One site was arable and cropped with second-year potato in 2015, while the other site had second-year rotational grass; the land use at both sites was identical to that in 2008-2009. These two sites are referred to as *AR1* and *RG1*, respectively. Land use treatments were replicated by selecting a second site for each land use category, subsequently referred to as *AR2* and *RG2*, respectively. The AR2 site was located 4.6 km to the west of, and *RG2* 1.7 km to the east of the paired *AR1-RG1* sites. Distribution and overviews of the four sites are presented in Figure S1.

### 2.2 Experimental design

In January 2015, an area of 10 m × 24 m was defined and sampled in 24 positions to ascertain peat depth. Sampling positions were georeferenced using a Topcon HiPer SR geopositioning system (Livermore, CA). On 25 February 2015, each site was fenced and three 10 m × 8 m experimental blocks defined (see Figure 1). Each area was further divided along its longitudinal axis to establish two 5 m × 24 m subplots.

For monitoring of water table (WT) depth, piezometer tubes (Rotek A/S, Sdr. Felding, Denmark) were installed to 150 cm depth at the centre of each block. On either side of the piezometers, at 2.7 m distance, collars of white PVC (base area: 55 cm × 55 cm, height: 12 cm [*RG*] or 24 cm [*AR*]) were installed to between 5 and 10 cm depth. In Figure 1, the sampling positions are referred to as S1 to S6. A 4-cm wide flange to support chambers extended outwards 2 cm



from the top. The support was fixed to the ground by four 40 cm pegs. Platforms (60 cm × 100 cm) of PVC were placed
beside each collar to prevent soil disturbance during gas sampling. The exact headspace of each collar was determined
from 16 individual measurements of distance from the upper rim; this procedure was repeated whenever collars were
removed and reinstalled in order to facilitate field operations.

Sets of diffusion probes for soil gas sampling were installed vertically within 0.5 m of the sampling positions S3-
S6 at sites *AR1* and *RG1*, while at *AR2* and *RG2* diffusion probes were only installed at S3 and S4. Gas sampling
positions were at 5, 10, 20, 50 and 100 cm depth. The stainless steel probes were constructed as described by Petersen
(2014): with a 10 cm$^3$ cell connected to the surrounding soil via a 3 mm diameter opening at the sampling depth that
was covered by a silicone membrane, and connected to the soil surface via two 18G steel tubes with Luer Lock fittings
(Petersen, 2014).

A HOBO Pendant Temperature Data Logger (HOBO, U.S.) was installed at 5 cm depth in block 2 at each site. A
mobile weather station (Kestrel 4500; Nielsen-Kellerman) was mounted at 170 cm height at site *RG1* for hourly
recording of air temperature, barometric pressure, wind speed and direction and relative humidity. Precipitation could
not be recorded on-site and was obtained from a meteorological station at Tylstrup (distance: 8 km), from where data to
fill a gap in air temperature was also obtained. Figure 3 (and subsequent Figures) shows air temperature and
precipitation over the experimental period.

**2.3 Management**

Management practices (fertiliser application, grass cuts and removal, potato harvest and soil tillage) of the farmers
outside the fenced experimental blocks were replicated within the blocks. One exception to this was fertilisation, where
only half of the blocks received N fertiliser (see below). Prior to tillage, cuts or harvest, the upper portion of the
piezometer tubes was removed (at *AR* sites to 50 cm depth) and the remaining part capped.

One subplot of the *RG1* site received, on 16 April (DOY105), 350 kg ha$^{-1}$ NS 27-4 fertiliser, corresponding to 94.5 kg
N ha$^{-1}$. Site *RG2* was fertilised with 20-25 t acidified cattle slurry (pH 6) on 5 May (DOY124), and again on 2 July
(DOY182), corresponding to 90-110 kg N ha$^{-1}$. On 2 July (DOY182), the application of acidified cattle slurry was
repeated, and a further 50 kg N ha$^{-1}$ was applied as NS 27-4 pelleted fertiliser; by accident this was given to the entire
field plot. The *AR1* site received 100 kg N ha$^{-1}$ as liquid NPS 20-3-3 fertiliser on 21 May (DOY140), while the *AR2* site
received 110 kg N ha$^{-1}$ as NS 21-24 pelleted fertiliser on 30 April (DOY119). According to a fertiliser database, NS
fertilisers contain equal amounts of ammonium (NH$_4^+$) and nitrate (NO$_3^-$), while NPS fertiliser is mainly NH$_4^+$.

At the *RG1* site, the grass was cut in late August, while at the *RG2* site this occurred in late June and on 9 September
(DOY251). Potato harvest at the *AR1* site took place in mid September, with interruptions due to significant rainfall. At
the *AR2* site, the potato harvest occurred on 23 September (DOY265).

The timing of field operations is shown together with rainfall and air temperature in Figures 3-6.





### 2.4 Field campaigns

A monitoring program was conducted in the spring 2015 from 3 March (DOY61) to 16 June (DOY166), and in autumn from 3 September (DOY245) to 10 November (DOY313). Weekly measurement campaigns were conducted at each of the four sites insofar as field operations permitted. During spring there were 14, 12, 14 and 15 weekly campaigns at the *RG1*, *AR1*, *RG2*, and *AR2* sites, respectively, the differences being due to interruptions for field operations. During autumn, there were 10, 10, 7 and 10 weekly campaigns at the *RG1*, *AR1*, *RG2*, and *AR2* sites, respectively. Two sites were visited during each field trip, either *AR1 + RG1* or *AR2 + RG2*.

With few exceptions each campaign was initiated between 9:00 and 12:00; the order of sites covered in each trip alternated from week to week. Campaigns included registration of weather conditions and WT depth, soil sampling, soil gas sampling, and $N_2O$ flux measurements.

### 2.4.1 Climatic conditions

Air temperature, relative humidity and barometric pressure were recorded upon arrival. Then WT depth was determined in each of the three piezometers (Figure 1). At *AR1* and *AR2*, WT depth in block 3 was further recorded at 30-minute time resolution for a period during autumn using MaT Level2000 data loggers (MadgeTech; Warner, NH).

Soil temperature at 5, 10 and 30 cm depth was measured within each block using a high precision thermometer (GMH3710, Omega Newport, Deckenpfronn, Germany), thus extending the continuous measurements of soil temperature at 5 cm depth mentioned earlier.

### 2.4.2 Soil sampling

Soil samples were collected separately from fertilised and unfertilised subplots by random sampling of six 20 mm diameter cores to 50 cm depth. Each core was split into 0-25 and 25-50 cm depth and the six samples from each depth pooled. The pooled samples were transported back to the laboratory in a cooling box for later analysis of mineral N and gravimetric water content.

On 23 April 2015 (DOY112), and again on 2 September (DOY244), intact cores (50 mm diameter, 300 mm length) were collected within 1 m distance from each of the six collars at *RG1* and *AR1* using a stainless steel corer (04.15 SA/SB liner sampler, Eijkelkamp, Giesbeek, Netherlands) equipped with a transparent plastic sleeve. The cylinder's lower end was capped with a 4 cm long cutting head, and hence sampling depths were 0 to 30 cm, 34 to 64 cm and 68 to 98 cm. The intact cores were capped and sealed, and transported in a cooling box to the laboratory, where they were stored at -20°C.

### 2.4.3 Soil gas sampling

Soil gas samples were taken in 6 mL pre-evacuated Exetainers (Labco Ltd, Lampeter, UK) as described by Petersen (2014). In brief, the diffusion probes were flushed with 10 mL $N_2$ containing 50 µL L$^{-1}$ ethylene (AGA, Enköbing,



Sweden) as a tracer using a plastic syringe. A three-way valve, mounted on the outlet tube, was fitted with a 10 mL glass syringe and an Exetainer. The displaced gas was quantatively collected in the glass syringe from where the soil gas sample, now partly diluted by the flushing gas, was transferred to the Exetainer. Finally, the probe was flushed with

$2 \times 60$ mL $N_2$ to remove ethylene, and Luer Lock fittings were capped. The $N_2$/ethylene gas mixture was also transferred directly to Exetainers ($n = 3$) as reference for gas chromatographic analysis. The number of soil gas samplings was often less than the number of flux measurements, partly because equipment had to be removed in periods with field operations.

### 2.4.4 Nitrous oxide flux measurements

Gas fluxes were measured with static chambers (60 cm $\times$ 60 cm $\times$ 40 cm) constructed from 4-mm white PVC and equipped with a rubber gasket (Emka Type 1011-34; Megatrade, Hvidovre, Denmark). Chambers were further equipped with a 12V fan (RS Components, Copenhagen, Denmark) for headspace mixing that was connected to an external battery (Yuasa Battery Inc.; Laureldale, PA, USA), a vent tube with outlet near the ground to minimize effects of wind (Conen and Smith, 1998; Hutchinson and Mosier, 1981), a temperature sensor (Conrad Electronic SE; Hirschau,

Germany), and a butyl rubber septum on top of each chamber for gas sampling. Handles attached to the top were used for straps fixing the chamber firmly against the collar. Gas samples (10 mL) were taken with a syringe and hypodermic needle immediately after chamber deployment, and then 15, 30, 45 and 60 minutes after closure. Gas samples were collected in 6 mL pre-evacuated Exetainer vials.

### 2.4.5 Soil analyses

Soil samples collected during the weekly campaigns were sieved and subsampled for determination of soil mineral N and gravimetric water content. Approximately 10 g fresh wt. soil was mixed with 40 mL 1 M potassium chloride (KCl) and shaken for 30 min. Concentrations of ammonium ($NH_4^+$) and nitrite ($NO_2^-$) + nitrate ($NO_3^-$) in filtered extracts were determined by autoanalyzer (Model 3; Bran+Luebbe GmbH, Norderstedt, Germany) using standard colorimetric methods (Keeney and Nelson, 1982). Gravimetric soil water content was determined after drying at 80°C for 48 hours.

Additional soil characteristics were determined on the intact soil cores collected in April and September at *AR1* and *RG1*. Five cm sections were subsampled from selected depths and analysed for water content, pH, electrical conductivity (EC), total soil organic C and total N and $NO_2^-$. The subsamples were further analysed for concentrations of total reactive Fe (TRFe), and soil from selected depth intervals from the September sampling were analysed for acid volatile sulfur (AVS) from, mainly, monosulfides, and for chromium reducible sulfur (CRS) derived from pyrite and

elemental S (Praharaj and Fortin, 2004). Soil organic C and N was further determined in bulk soil samples collected in the same weeks at *RG2* and *AR2*.

Soil pH and EC were measured with a Cyberscan PC300 (Eutech Instruments; Singapore). Total soil organic C and total N were measured by high temperature combustion with subsequent gas analysis using a vario MAX cube CN analyser (Elementar Analysensysteme GmbH; Langenselbold, Germany). Soil $NO_2^-$-N concentrations in soil:water

extracts (1:5, w/v) were determined by a modified Griess-Ilosvay method (Keeney and Nelson, 1982).




The analysis of TRFe was done using a dithionite-citrate extraction (Carter and Gregorich, 2007; Thamdrup et al., 1994) followed by $Fe^{2+}$ analysis by the colorimetric ferrozine method including hydroxylamine as reducing agent (Viollier et al., 2000). The extraction dissolves all free (ferric) Fe oxides except magnetite ($Fe_3O_4$). It also dissolves (ferrous) Fe in iron monosulfide (FeS), but not pyrite ($FeS_2$).

Quantification of AVS and CRS was based on passive distillation adapted from Ulrich et al. (1997) and Burton et al. (2008). Briefly, 0.5 g soil and a trap with 4 mL alkaline $Zn^{2+}$ solution (5%) was placed in 120 mL butyl-stoppered (and crimp-sealed) serum bottles, which were repeatedly (3 ×) evacuated (0.1 kPa) and pressurized with $N_2$ (150 kPa) to remove $O_2$, eventually leaving the headspace at atmospheric pressure with $N_2$. Acid volatile sulfide (primarily FeS) was liberated and trapped after injection of 12 mL anoxic 2 M HCl followed by sonication (0.5 h) and incubation (24 h) on a

rotary shaker (20ºC). Using the same approach with replicate soil samples, combined AVS and CRS (primarily S˚ and $FeS_2$) was trapped after injection of 12 mL 1 M $Cr^{2+}$ in 2 M HCl, prepared by reduction of $CrCl_3$ (Røy et al., 2014). Sulfide (ZnS) in the two traps was measured colorimetrically using diamine reagent (Cline, 1969), and CRS was then calculated by difference.

### 2.4.6 Gas analyses

Nitrous oxide concentrations were analysed on an Agilent 7890 gas chromatograph (GC) with a CTC CombiPal auto-sampler (Agilent, Nærum, Denmark). The instrument had a 2 m back-flushed pre-column with Hayesep P connected to a 2 m main column with Poropak Q. From the main column, gas entered an electron capture detector (ECD). The carrier was $N_2$ at a flow rate of 45 mL $min^{-1}$, and Ar-$CH_4$ (95%/5%) at 40 mL $min^{-1}$ was used as make-up gas. Temperatures of the injection port, columns and ECD were 80, 80 and 325°C, respectively. Concentrations were quantified with

reference to synthetic air and a calibration mixture containing 2013 μL $L^{-1}$ $N_2O$. Soil profile $N_2O$ concentrations were frequently at several hundred μL $L^{-1}$; linearity of the EC detector was ascertained for the range 0.3-50 μL $L^{-1}$, but this may not have been the case across the entire range observed, and therefore the reported concentrations outside 0-50 μL $L^{-1}$ are uncertain.

Ethylene concentrations in soil gas samples and flushing gas were analysed following a separate injection with an

extended run time. All GC settings were as described earlier, except that gas from the main column was directed to a FID detector supplied with 45 mL $min^{-1}$ $H_2$, 450 mL $min^{-1}$ air, and 20 mL $min^{-1}$ $N_2$, and with an operational temperature of 200°C.

### 2.5 Data processing and statistical analyses

Nitrous oxide ($N_2O$) fluxes were calculated in R (version 3.2.5, R Core Team, 2016) using the package HMR (Pedersen

et al., 2010). This program analyses non-linear concentration-time series with a regression-based extension of the model of Hutchinson and Mosier (1981), and linear concentration-time series by linear regression (Pedersen et al., 2010). Statistical data ($p$ value, 95% confidence limits) are provided for both categories of fluxes. The choice to use a linear or non-linear flux model was made based on scatter plots and the statistical output.




Tests for normality (Shapiro-Wilk) showed that $N_2O$ fluxes were skewed, and the data were therefore log transformed prior to statistical analyses comparing $N_2O$ fluxes on individual sampling days, as well as cumulative fluxes during spring and autumn. To meet assumptions of homoscedasticity and residual-normality, the $N_2O$ fluxes for each sampling day, and for each combination of crop and fertilisation regime, were modelled using a generalised linear mixed model defined with the identity link function, the gamma distribution, and Gaussian random components. The models contained a fixed effect representing an interaction between crop, fertilisation and observation day, and random

effects representing site and the experimental unit for which the repeated measures were performed.

    Cumulative $N_2O$ emissions were estimated by the method of trapezoidal approximation of the integral of the emission curve by defining specially constructed contrasts using the glmer function of the lme4 package (Duan et al., 2017).

    Inter-dependencies of several variables with a potential to regulate fluxes of $N_2O$ were studied using graphical

models (Jørgensen and Labouriau, 2012; Whittaker, 1990). Explanatory variables included soil temperature at 5 cm depth (Temp5, representing top soil), soil temperature at 30 cm depth (Temp30, representing soil near the capillary fringe), $NH_4^+$ and $NO_3^-$ concentrations in the top 25 cm of the soil profile (AmmoniumT and NitrateT, representing nutrient status in the top soil), and finally $N_2O$ concentration in the soil gas diffusion probe closest to, but above the WT depth ($N_2O_{WT}$, representing N transformations in the capillary fringe). In graphical models, the variables are represented

in an undirected graph by a set of vertices (points) and edges (lines connecting points). Two vertices are connected by an edge when the conditional correlation of the two corresponding variables, given all the other variables, is different from zero. Such connections show that the two variables carry information on each other that is not already contained in the other variables. Moreover, the absence of an edge connecting two vertices indicates that (even a possible) association between the two corresponding variables can be entirely explained by the other variables. A separate

analysis was conducted for each combination of season and crop. The graphical models were inferred by finding the graphical model that minimised the Bayesian information criterion as implemented in the R package gRapHD (Abreu et al., 2010); this inference procedure has optimal properties (Haughton, 1988).

**3 Results**

**3.1 Climatic conditions**

During the spring monitoring period, the daily mean air temperature varied between 1 and 15°C, with an increasing trend over the period, and total rainfall was 220 mm. During the autumn monitoring period, the daily mean air temperature declined from 15 to 5°C, and total rainfall was 148 mm; the most intense daily rainfall events during spring and autumn were 16.9 and 33.2 mm, respectively. For 2015 as a whole, the annual mean air temperature in the area was 8.7°C, and annual precipitation 920 mm. Daily temperature and precipitation during the monitoring periods are

presented in connection with Figure 3-6 in order to align this information with $N_2O$ fluxes and other supporting information.



Soil temperature was recorded at 5, 10 and 30 cm depths during the sampling campaigns. There was a strong diurnal trend in hourly soil temperatures at 5 cm depth at each of the four sites (Figure S2).

### 3.2 Soil characteristics

Selected site characteristics are shown in Table 1. The results from *RG1* and *AR1* represent mean and standard error of intact soil cores ($n = 6$) collected on 23 April; results from *RG2* and *AR2* were based on soil sampled in campaigns during the same week. The soil at all sites was acidic, with pH ranging from 4.6 to 5.5. At the paired sites *AR1* and *RG1* a decline in pH was indicated at 40-50 cm depth (Table 1). Across all sites electrical conductivity (EC) ranged from 32 to 107 $\mu$S m$^{-1}$; at *AR1* and *RG1* the lowest values were observed at 93-98 cm depth where the soil profile was

dominated by sand.

Soil organic C and N$_{tot}$ reflected peat depth (Table 1). At the paired sites *AR1* and *RG1*, some mixing of peat with the underlying sand was indicated at 47.5-52.5 cm depth, whereas SOC at *RG2* only just met the requirements for being defined as an organic soil. Site *AR2* was characterised by a uniform peat layer extending below the lowest sampling depth (98 cm). The C:N ratios varied from 14 to 69, but were constant in peat layers. The C:N ratio was close to 20 at

*AR1* and *RG1*, around 15 at *RG2*, and around 25 at *AR2* (Table 1).

Total reactive Fe (TRFe) in soil profiles from *AR1* and *RG1* ranged from 0.18 to 4.99 mg g$^{-1}$ dry wt. soil; at both sites TRFe declined below 20 cm depth and was close to zero in the sand below the peat layer (Table 1). Water table depth at sampling on 23 April (DOY112) was at 70-84 cm, and hence TRFe concentrations also declined in the capillary fringe. Concentrations of reactive Fe were two orders of magnitude higher than concentrations of iron sulfides

at these sites. Acid volatile S ranged from 1.65 to 3.50 $\mu$g S g$^{-1}$ soil but there was no clear relationship with soil depth (Table 1). Chromium reducible S contents decreased with depth from 146 to 40 $\mu$g S g$^{-1}$ dry wt. soil, and from 128 to 48 $\mu$g S g$^{-1}$ dry wt. soil at sites *RG1* and *AR1*, respectively (Table 1).

The concentrations of TRFe in intact soil cores collected on 23 April and 2 September 2015 are shown in Figures 2B (*RG1*) and 2D (*AR1*). The highest concentrations at both sites occurred at 20 cm depth, declining to near zero at *c.* 1

m depth. There was little difference in the distribution of TRFe between April and September, except that the profile from *AR1* indicated a sink for TRFe at 40-60 cm depth (Figure 2D). There was a strong relationship between TRFe and SOC across all sites (r$^2$ = 0.78, $n$ = 16).

### 3.3 Soil mineral N dynamics

Soil concentrations of ammonium (NH$_4^+$) and nitrate (NO$_3^-$) at 0-25 and 25-50 cm depth on sampling days are included

as Supplementary Information in Tables S1-S4. At the arable sites there was an accumulation of mineral N at both depth intervals during May (Table S2, S4); the accumulation at *AR1* was much greater than at *AR2* and was recovered as both NH$_4^+$ and NO$_3^-$, whereas at *AR2* only NO$_3^-$ -N accumulated.



Fertilisation resulted in dramatic increases in $NH_4^+$-N and $NO_3^-$-N concentrations to 100-200 µg g$^{-1}$ dry wt. soil at all sites except *RG2* where acidified cattle slurry was surface applied; it is not clear if the slurry infiltrated to greater

depth, or if plant uptake was very effective. The residence time for mineral N was generally longer at *AR* compared to *RG* sites, presumably because of N uptake by the grass sward. There was some accumulation of $NO_3^-$ in the weeks after fertilisation at all sites, and also transport to 25-50 cm depth.

As stated above, $NO_2^-$-N and $NO_3^-$-N were not analysed separately in soil from the weekly sampling campaigns, but $NO_2^-$-N concentrations were determined in the intact profiles collected from *RG1* and *AR1* on 23 April and 2

September 2015. These results are shown in Figures 2A (site *RG1*) and 2C (*AR1*). In April, the concentration of $NO_2^-$-N at both sites was highest (*c.* 10 µg g$^{-1}$ dry wt. soil) around 40 cm depth and declined towards the surface and deeper layers; there was a notable decline in $NO_2^-$-N at 50 cm depth in *AR1* (Figure 2C), where also a sink for TRFe was indicated. In August, $NO_2^-$-N concentrations were < 1 *µ*g g$^{-1}$ dry wt. soil at both sites.

**3.4 Groundwater table dynamics**

In order to facilitate data interpretation, weather data have been aligned with $N_2O$ fluxes, soil $N_2O$ concentration profiles and WT dynamics of fertilised and non-fertilised subplots of sites *RG1* and *RG2* in spring (Figure 3), sites *AR1* and *AR2* in spring (Figure 4), sites *RG1* and *RG2* in autumn (Figure 5), and sites *AR1* and *AR2* in autumn (Figure 6). The results are presented in this and the following sections.

The paired sites *RG1* and *AR1* were within fields that had a small slope (*c.* -0.5º) from North to South, which

resulted in WT depths being around 10 cm lower in Block 1 compared to Block 3; the average WT depths at sampling are presented as grey lines overlying the contour plots in Figures 3 and 4. During spring, WT depth at the paired sites *RG1* and *AR1* ranged from 17 to 81 cm, with a steady decline until the end of April (DOY120) that was followed by a period with fluctuations around 60-80 cm depth due to frequent rainfall. During the first half of September (DOY243 to 258), rainfall caused the WT to rise from 80 to 40 cm depth. The continuous measurements of WT depth (data not

shown) revealed, however, that on two occasions (DOY247 and 259) the WT depth rose to 20 cm depth and only declined gradually during the following days. From mid September followed a period with a gradual decline in WT depth until early November where WT rose from 90 to 45 cm depth during a period with intense rainfall.

At *RG2*, the WT depth was mostly at 50-60 cm depth during spring, but rose temporarily to 30 cm depth by DOY154, i.e., 3 June. In the autumn, sampling campaigns could not be initiated until DOY259 due to harvest. By this

time the WT was close to the surface following intense rainfall, but then declined rapidly in the sandy subsoil.

The WT depth at site *AR2* varied between 45 and 60 cm depth during spring except for a transient increase to 34 cm depth in early June. During autumn, the WT depth rose to the soil surface on two occasions in September (DOY248 and DOY259), and then gradually withdrew until early November when rainfall caused a dramatic increase, as also observed at sites *RG1* and *AR1*.

**3.5 Soil profile $N_2O$ concentrations**



Soil $N_2O$ concentrations (or equivalent gas phase concentrations in saturated parts of the profile) are presented as contour plots; the actual concentrations of $N_2O$ as determined with passive diffusion samplers are presented in Tables S5 (spring) and S6 (autumn).

Under the rotational grass at site *RG1*, soil $N_2O$ concentrations during spring were mostly between 0.1 and 3 $\mu$L L$^{-1}$. A higher concentration (15 $\mu$L L$^{-1}$) was observed at 40-80 cm depth in the fertilised subplot around DOY135, but only in the lower end of the field plot. At *RG2* concentrations of $N_2O$ in the soil during spring were generally similar to those at *RG1*, although there were more values in the 1-10 $\mu$L L$^{-1}$ concentration range. However, on 4 June (DOY155) a dramatic increase in $N_2O$ concentration occurred in the fertilised part of the plot with a maximum of 560 $\mu$L L$^{-1}$ at 100 cm depth. This followed a rise in WT depth as a result of rainfall. Soil $N_2O$ concentrations in the unfertilised plot also increased around this time, but mainly near the soil surface. In both parts of the field plots the increase occurred below the WT depth.

During autumn, $N_2O$ concentrations in the soil profile at the *RG1* and *RG2* sites both varied between 0 and 12 $\mu$L L$^{-1}$ independent of fertilisation; there was a tendency for higher emissions at 10-20 cm depth.

The arable site at *AR1*, with sampling positions located 10-20 m from those of site *RG1*, showed very different soil $N_2O$ concentration dynamics. There was a consistent and dramatic accumulation of $N_2O$ at 50 and 100 cm depth, and concentrations during spring averaged 340 and 424 $\mu$L L$^{-1}$, respectively. In contrast, at 5, 10 and 20 cm depth the average $N_2O$ concentrations were 10-30 $\mu$L L$^{-1}$, and there was no clear response to fertilisation on DOY140 in terms of soil $N_2O$ accumulation. There was within-site heterogeneity in soil conditions, as the highest concentrations were observed in the part of the field plot without fertilisation. Between DOY75 and 100 the concentrations of $N_2O$ at 50 cm depth were 2-3 fold higher than at 100 cm depth, indicating $N_2O$ production in the capillary fringe. At site *AR2,* the highest $N_2O$ concentrations during spring were consistently observed at 20 cm depth, but gradually declining to reach the background level of 0.3 $\mu$L L$^{-1}$ in mid May (DOY131). In the unfertilised field plot, the $N_2O$ concentration then increased again at 20 cm depth to reach 272 $\mu$L L$^{-1}$ following rainfall and the WT rising to 35 cm depth. With fertilisation, soil $N_2O$ concentrations were even higher at 10 cm depth and reached 386 $\mu$L L$^{-1}$ at the last sampling in mid June.

September was characterised by heavy rainfall, and at site *AR1* a substantial rise in the WT from 80 to 40 cm depth was observed. Soil $N_2O$ concentrations showed a dual pattern, with maxima at 10 and 100 cm depth through to DOY265 (end of September), by which time soil $N_2O$ rapidly declined as the WT withdrew. Nitrous oxide concentrations equivalent to several hundred $\mu$L L$^{-1}$ were measured even at 5 cm depth during this period. During late autumn, the $N_2O$ concentration at 0-50 cm depth varied between 0 and 20 $\mu$L L$^{-1}$, whereas at 100 cm depth it remained high at 100-850 $\mu$L L$^{-1}$. At site *AR2,* the groundwater level was higher than at *AR1* and came close to the soil surface by mid September. Soil $N_2O$ accumulation upon saturation of the soil took place in both fertilised and unfertilised plots, again with the highest concentrations at 20 cm depth. A secondary increase was observed at the last sampling on DOY313 in November, in response to a period with rainfall and a rapid WT rise.





**3.6 Nitrous oxide fluxes**

Nitrous oxide fluxes during spring are shown in Figure 3 for sites *RG1* and *RG2*, and in Figure 4 for sites *AR1* and *AR2*. The corresponding results from autumn are shown in Figures 5 and 6. In each case results for fertilised and unfertilised subplots are shown separately.

Spring soil $N_2O$ fluxes at *RG1* ranged from 0 to 550 µg $N_2O$ $m^{-2}$ $h^{-1}$, with no effect of fertiliser amendment. The
grass sward showed a strong response to fertilisation (not shown), and presumably there was a rapid uptake of the N added. At site *RG2*, however, a peak in $N_2O$ flux occurred on DOY153, and the flux was still elevated at the next two samplings. This high flux coincided with elevated soil profile $N_2O$ concentrations, as described above.

At site *AR1* the $N_2O$ fluxes were generally higher than at the *RG1* site. Notably, fluxes during early spring of 2000-6000 µg $N_2O$ $m^{-2}$ $h^{-1}$ were higher than in late spring where again no effect of N fertilisation on DOY140 was observed.
Hence, the higher emissions were associated with soil conditions and not fertilisation. The reference potato field at site *AR2* showed a different pattern, with $N_2O$ fluxes remaining low during early spring and for several weeks after fertilisation on DOY119). The highest observations were, independent of fertilisation, occurring in June when a WT rise to 30 cm depth was observed.

In the autumn, $N_2O$ fluxes from site *RG1* were consistently low. The first sampling at site *RG2* was on DOY257 in
mid September, where elevated $N_2O$ fluxes of 1000-2000 µg $N_2O$ $m^{-2}$ $h^{-1}$ were seen, dropping within 1-2 weeks to near zero flux.

Nitrous oxide fluxes at site *AR1* were high at 4000-10,000 µg $N_2O$ $m^{-2}$ $h^{-1}$ during September independently of N fertilisation, and subsequently declined to near zero. The high fluxes coincided with a rise in WT from 80 to 40 cm depth, and the decline in fluxes with WT withdrawal. At site *AR2* the pattern in $N_2O$ fluxes was similar, and again the
dynamics in $N_2O$ flux reflected WT dynamics.

Cumulative $N_2O$ fluxes were calculated for the 90-98 d monitoring period in spring and 47-71 d period in autumn (Table 2). At *RG* sites, the average $N_2O$ flux from fertilised grassland was significantly higher than from unfertilised grass (7.3 *vs.* 2.0 kg $N_2O$ $ha^{-1}$) during spring. At *AR* sites with potato there was no significant effect of N fertilisation, but much higher overall $N_2O$ emissions of 15-17 kg $N_2O$ $ha^{-1}$ occurred when compared to *RG* sites. In autumn there
were no residual effects of N fertiliser application in spring, and overall average emissions of around 2 and 15 kg $N_2O$ $ha^{-1}$ were observed at *RG* and *AR*, respectively.

**3.7 Interrelationships between driving variables of $N_2O$ production**

Graphical models were used to explore the dependence structure among selected soil variables and $N_2O$ fluxes. At grassland sites in spring (Figure 7A) and autumn (Figure 7B), and at the arable sites in spring (Figure 7C), the only
variable with a direct link to $N_2O$ flux was soil $N_2O$ concentration above the WT. For example, in the analysis of *AR* sites in spring the variables $N_2O$ flux and Temp5 were not directly connected, which means that any correlation between Temp5 and $N_2O$ flux could be completely explained by the other variables. Furthermore, the fact that the



variable $N_2O_{WT}$ separated the variable $N_2O$ flux from Temp5 (and also from all the other variables) indicated that any information that Temp5 might contain on $N_2O$ flux was completely contained in the variable $N_2O_{WT}$. The only

exception to this pattern was for *AR* sites in autumn (Figure 7D), since here the variables with a direct bearing on $N_2O$ flux were Temp30 and NitrateT ($NO_3^-$-N concentration in the top soil).

## 4 Discussion

Spring and autumn monitoring periods together covered 152-174 d, and during these periods total emissions were 3-6 kg $N_2O$-N ha$^{-1}$ for rotational grass, and 19-21 kg $N_2O$-N ha$^{-1}$ for arable sites with a potato crop. This may be compared

with annual emissions corresponding to 24 and 61 kg $N_2O$-N ha$^{-1}$, respectively, which were reported for the same area and land uses by Petersen et al. (2012). The observed $N_2O$ emissions were at the high end (*RG*) or clearly above (*AR*) the IPCC emission factors for drained organic soil (IPCC, 2013) of 8 and 13 kg $N_2O$-N ha$^{-1}$ yr$^{-1}$ for nutrient rich grassland and cropland, respectively. They were also higher than averages reported for grassland and cropland in a recent meta-analysis of emissions from organic soils in Europe (Leppelt et al., 2014), which were 6 and 10 kg $N_2O$-N

ha$^{-1}$ yr$^{-1}$, respectively.

Leppelt et al. (2014) concluded that, across Europe, high $N_2O$ emissions from arable organic soil were associated with croplands having a pH below 4.7, C:N ratios below 30-35, and WT depths of 0.2-0.9 m, and they found a significant positive relationship with annual precipitation. This was based on statistical relationships across a wide range of soil types, land management practices and average annual conditions, and thus specific mechanisms behind

$N_2O$ emissions could not be derived. Here, we investigated seasonal dynamics of $N_2O$ emissions and soil conditions in a specific area (Store Vildmose) which was identified by Leppelt et al. (2014) as a hotspot for $N_2O$ emissions, with an expectation that higher spatial and temporal resolution could help understand environmental controls and possible mechanisms behind the high emissions of $N_2O$.

### 4.1 Water table depth and season

It is well established that $N_2O$ emissions from organic soil may be enhanced by drainage (Martikainen et al., 1993; Taft et al., 2017). The response will appear within days, as shown by Aerts and Ludwig (1997) in an incubation study with an oscillating WT, where lowering the WT enhanced $N_2O$ emissions repeatedly (Aerts and Ludwig, 1997). Similarly, a stimulation of $N_2O$ emissions was observed by Goldberg et al. (2010) when simulating drought under field conditions, but a pulse of $N_2O$ also followed rewetting. In the present study, the response to WT drawdown was complex, i.e., at

sites *RG1* and *AR1* a stimulation of $N_2O$ emissions was observed as WT declined in early spring, while this was not evident at sites *RG2* and *AR2*, and during autumn there was generally no effect of WT drawdown on $N_2O$ emissions. In contrast, rising WT and/or increasing soil wetness in late spring and autumn resulted in a consistent increase in $N_2O$ emissions at all sites. Hence, the relationship between WT depth and $N_2O$ emission showed seasonal patterns and site-specific effects suggesting that land use and soil properties also influenced the potential for $N_2O$ emissions.

### 4.2 Nutrient status and land use





The repeated increase in $N_2O$ emissions after WT drawdown reported by Aerts and Ludwig (1997) was observed only with eutrophic peat, while a mesotrophic peat showed no effect of WT treatment on $N_2O$ emissions, which were consistently low. A similar interaction between nutrient status and WT depth has been observed in field studies comparing $N_2O$ emissions from minerotrophic and ombrotrophic boreal peatlands (Martikainen et al., 1993; Regina et al., 1996). Thus, nutrient status, and probably N availability in particular, was important for the $N_2O$ emission potential at *AR* sites used for potatoes. The *RG* sites with rotational grass, in contrast, showed much lower $N_2O$ emissions compared to *AR* despite similar soil conditions and N fertiliser input.

Grasslands on organic soil will generally show lower emissions of $N_2O$ compared to arable organic soil (Eickenscheidt et al., 2015; Petersen et al., 2012), presumably because plants compete successfully with microorganisms for available N. Schothorst (1977) estimated peat decomposition indirectly from the N-content of herbage yield of grassland and concluded that the soil supplied 96 kg N ha$^{-1}$ at a drainage depth of 25 cm, but 160 and 224 kg N ha$^{-1}$ with the WT in drainage ditches at 70 and 80 cm depth, respectively. This does indicate that the grass sward competes effectively for N mineralised from soil organic matter at the WT depths observed in the present study. Maljanen et al. (2003) found little response to fertilisation during a two-year study. Regina et al. (2004) also found little effect of fertilisation, but did observe a peak in $N_2O$ emissions in late spring after rainfall. In the present study there was also no immediate response to input of N in fertilisers until to rainfall events, and hence the relationship between nutrient availability and $N_2O$ emissions is complex.

### 4.3 Nitrogen dynamics and $N_2O$ in soil profiles

Only pooled soil samples from 0-25 and 25-50 cm depth were available for mineral N analyses (Table S1-S4). However, information about N transformations can also be derived from soil $N_2O$ concentration profiles (Goldberg et al., 2008). The soil gas diffusion probes used in this study were installed vertically and thus did not disturb soil stratification prior to monitoring. At *RG* sites, soil $N_2O$ concentrations were generally low and did not provide clear evidence for N mineralisation from the peat. In contrast, during spring at site *AR1*, there was a dramatic accumulation of $N_2O$ at 50 and 100 cm depth, whereas at *AR2* the highest concentrations observed were lower and peaked at 20 cm depth, in accordance with a higher groundwater table. This implies that peat decomposition was a significant source of mineral N around the WT depth, and that biotic or abiotic processes resulted in extensive $N_2O$ formation. At site *RG2,* a dramatic accumulation of $N_2O$ at 1 m depth in late May indicated that mineral N from the acidified cattle slurry had leached from the top soil (Figure 3), consistent with the fact that mineral N concentrations were never elevated in the top soil (Table S3). Effects of N fertilisation on $N_2O$ emissions were thus observed, but only as interacting effects of rising WT and/or an increase in wetness of the unsaturated soil.

September 2015 was characterised by heavy rainfall in the Store Vildmose district, and the rapid rise in the WT toward the soil surface resulted in accumulation of $N_2O$ in the soil at all sites, but $N_2O$ concentrations ranged from around 10 µL L$^{-1}$ at *RG* to several hundred µL L$^{-1}$ at *AR* sites. This was accompanied by a near depletion of $NO_3^-$-N that was most likely derived from mineralisation of N in potato crop residues.

### 4.4 Environmental controls – main effects and interactions



There are at the continental scale significant effects of environmental controls across land use categories, nutrient status and climate when considering annual average $N_2O$ emissions (Leppelt et al., 2014; Mu et al., 2014). The previous sections have shown that these main effects are, when considering a higher spatial and temporal resolution, in fact modified by other variables. As a consequence, the main drivers of $N_2O$ emissions may not be constant across the year

or between different land uses, and it is relevant to consider what is the most important constraint on $N_2O$ emissions in each season.

Graphical models showed that the strongest predictor for $N_2O$ emissions from both grassland and arable soil in spring, and from grassland in autumn, was the equivalent soil $N_2O$ gas phase concentration near the groundwater table. The implication is that soil N transformations at depth in the soil, and not in the top soil, were the main source of $N_2O$

escaping to the atmosphere. This is in accordance with the study of Goldberg et al. (2010) showing that $N_2O$ emissions from a minerotrophic fen were produced at 30-50 cm depth. Peat decomposing in the capillary fringe as the WT declined could have resulted in $N_2O$ formation, and indeed the highest concentrations of $N_2O$ were observed at 50 cm depth, or 20 cm depth in the case of site *AR2* (Table S5). Also, there appeared to be a drain for TRFe and $NO_2^-$ at 50 cm depth in late April (Figure 2); possible mechanisms are discussed in section 4.5. However, it should also be noted that at

site *AR1* the $N_2O$ concentrations at 100 cm depth were also extremely high, in accordance with the observations reported by Petersen et al. (2012), and hence there were most likely additional processes forming $N_2O$ under strictly anaerobic conditions.

The graphical model analysis further revealed that at arable sites the regulation of $N_2O$ emissions in autumn was different from spring, since in the autumn period $NO_3^-$ in the top soil, together with temperature, were proximal controls

of $N_2O$ emissions, although it should be noted that accumulation of $NO_3^-$ was much greater at site *AR1* compared to *AR2*. This result is consistent with $NO_3^-$ accumulating in the soils following mineralisation and subsequent nitrification of N in residues from the potato crop. Rainfall most likely triggered denitrification at the arable sites, by increasing soil water-filled pore space and hence impeding oxygen supply, as observed in mineral soil (Barton et al., 2008). This was indicated by $N_2O$ concentrations increasing in the capillary fringe above the WT depth (Figure 6). In other cases the

accumulation of $N_2O$ evidently occurred below the WT depth, but whether this $N_2O$ was derived from residue N leaching to greater soil depth or from peat decomposition is not clear.

**4.5 Possible sources of $N_2O$**

**4.5.1 Spring**

The amount and vertical distribution of $N_2O$ within soil profiles differed between sites, land use categories and seasons,

indicating that different processes or drivers could be involved. Bacterial nitrification, denitrification, and nitrifier-denitrification, are typically considered the main sources of $N_2O$ (Braker and Conrad, 2011), but (aerobic) ammonia oxidising bacteria (AOB) are scarce in acid peat despite the presence of nitrite oxidising bacteria (Regina et al., 1996). Bacterial ammonia oxidation under anaerobic conditions (Anammox) has been observed in peat soil (Hu et al., 2011), but other studies indicate that in acid peat ammonia oxidising archaea (AOA) predominate both in abundance and

activity (Herrmann et al., 2012; Stopnišek et al., 2010). Stopnišek et al. (2010) found that AOA activity was not





stimulated by an external source of $NH_4^+$ and concluded that the activity was associated with N release from decomposing soil organic matter. Accordingly, peat decomposition could be a limiting factor for ammonia oxidation during spring, a limitation that was alleviated as the WT declined and oxygen entered deeper soil layers. Nitrite accumulation in the capillary fringe in late April (Figure 2) would be consistent with AOA activity. The study of

Herrmann et al. (2012) found a rapid response to drainage and rewetting in the levels of $NO_3^-$ accumulation and gene transcription, which confirms that nitrification activity in organic soil can react dynamically to WT changes.

The presence of AOA in the capillary fringe does not, however, account for the significant accumulation of $N_2O$ below the WT depth, which implied that ammonia oxidation also occurred in saturated peat layers, as a precondition for $N_2O$ formation. The sites investigated contained significant amounts of reactive iron (TRFe; cf. Table 1). The

concentrations of TRFe were much higher than concentrations of AVS, and thus iron oxides/hydroxides rather than iron sulfides were probably the source of TRFe, in accordance with the composition of bog iron reported by Madsen et al. (2000). A sorption capacity of 1.3 mmol (= 71.5 mg) ferric iron per gram humic acid isolated from a bog was reported by (Davies et al., 1997). Since humic acids constitute 50-90% of organic matter in peat (Perminova et al., 2003), and the concentration of TRFe was strongly related to SOC content in this study, ferric iron associated with peat is a potential

source of electron donor, and the process involved could thus be anaerobic ammonia oxidation coupled with ferric iron reduction, Feammox, a process which may lead to $NO_2^-$ formation below pH 6.5 (Yang et al., 2012):

$$6Fe(OH)_3 + 10H^+ + NH_4^+ \rightarrow 6Fe^{2+} + 16H_2O + NO_2^-$$

The accumulation of $NO_2^-$ at low pH would result in product inhibition from $HNO_2$ in the absence of a mechanism to remove $NO_2^-$ (van Cleemput and Samater, 1995). Chemodenitrification is an abiotic reaction of $NO_2^-$ or $NO_3^-$ with $Fe^{2+}$

that results in $N_2O$ formation (Jones et al., 2015), and thus it was a potential source of $N_2O$ in this study. The half-life of $NO_2^-$ declines with pH and increasing availability of $Fe^{2+}$ (Van Cleemput and Samater, 1996). Such a mechanism would explain the relative depletion of $NO_2^-$ and TRFe at 50 cm depth at site *AR1* (Figure 2C,D) compared to site *RG1* (Figure 2A,B), and the corresponding difference in $N_2O$ accumulation in early spring (cf. Figures 3 and 4).

We thus propose that AOA activity was responsible for $NO_2^-$ accumulation in the capillary fringe during spring,

while anaerobic ammonia oxidation, possibly *via* Feammox¸ predominated in the saturated zone. Pitcher et al. (2011) also reported such a vertical seggregation of AOA and Anammox bacteria between micro-aerophilic (1-2% of air saturation) and anaerobic conditions, albeit in a deep-sea water column. The accumulation of $NO_2^-$ above the WT depth may be explained by the fact that the capillary fringe is a mosaic of aerobic and anaerobic sites (cf. Estop-Aragonés et al., 2012) where, probably, $NO_2^-$ diffused between sites of nitrification and denitrification in response to concentration

gradients.

Nitrous oxide accumulated mainly around or below the WT depth (Figures 3-4), indicating that denitrification of fertiliser-derived $NO_3^-$ was not a major source of $N_2O$ in spring. As already stated, $NO_2^-$ reduction to $N_2O$ could be the result of chemodenitrification. Other possible mechanisms of $N_2O$ formation in acid soils include nitrifier-denitrification, bacterial or fungal heterotrophic denitrification with $NO_2^-$ or $NO_3^-$ as electron acceptor (Liu et al., 2014),

hybrid formation of $N_2O$ *via* abiotic codenitrification (Spott et al., 2011); and ammonia oxidation (Stieglmeier et al., 2014). Fungal denitrification is less sensitive to pH than bacterial denitrification (Herold et al., 2012), and here $N_2O$ is





the end product (Maeda et al., 2015). Stieglmeier et al. (2014) described an AOA isolated from soil that produced $N_2O$ at a rate of 0.09% of the $NO_2^-$ produced, and this production was largely independent of $O_2$ availability.

### 4.5.2 Autumn

The graphic model analysis indicated that $NO_3^-$ was important for the extreme $N_2O$ emissions from arable soil observed during (early) autumn. By this time the soil was well-aerated and $NO_2^-$ levels were low (Fig. 2), whereas accumulation of $NO_3^-$ was significant, especially at *AR1* (Table S2 and 4). Presumably conditions had been favourable for nitrification activity in the previous weeks. The rapid shift in WT depth towards the surface in early September, in connection with heavy rainfall, resulted in extremely high $N_2O$ emissions from potato fields irrespective of fertilisation,

and also from fertilised grasssland at site *RG2*. This could well be a result of intense denitrifier activity, as also indicated by soil $N_2O$ accumulation near the soil surface.

### 5 Conclusion

The hypotheses of this study were largely confirmed with $N_2O$ emissions from arable organic soil were extremely high, and clearly higher compared to grassland, independent of fertilisation. It was also confirmed that $N_2O$ emissions

coincided with WT drawdown (spring) or increase (autumn), and soil $N_2O$ concentration profiles indicated that $N_2O$ was to a large extent produced in the capillary fringe. Unexpectedly, however, there was evidence that the source of $N_2O$ in arable soil differed between spring and autumn. In the spring, there was no clear response to the input of $NO_3^-$ in fertiliser, and $N_2O$ emissions mainly reflected the dynamics of $N_2O$ accumulation near the WT. We propose that decomposing peat was the main source of $N_2O$ during WT drawdown, and that possibly a combination of Feammox and

chemodenitrification caused accumulation of $N_2O$. In contrast, the very high $N_2O$ emissions observed during the rapid rise in WT depth in early autumn were significantly related to $NO_3^-$ availability in the top soil. Reducing surplus N in the soil, for example by use of a plant cover, and stabilisation of WT depth during the year, appear to be keys to controlling $N_2O$ emissions.

### 6 Acknowledgements

This study received financial support from the Danish Research Council for the project "Sources of $N_2O$ in arable organic soil as revealed by $N_2O$ isotopomers" (DFF – 4005-00448). We would like to thank the dedicated staff involved in field campaigns, including Bodil Stensgaard, Søren Erik Nissen, Sandhya Karki, Kim Johansen, Karin Dyrberg, Holger Bak and Stig T. Rasmussen. We would also like to acknowledge the support of three farmers hosting the field sites: Poul-Erik Birkbak, Rasmus Christensen and Jørn Christiansen.





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





**Table 1.** Soil characteristics at the four monitoring sites in late April 2015 (AVS and CRS were measured from soil samples in September 2015).

| | Depth (cm) | pH | EC | SOC (g 100 g⁻¹) | N$_{tot}$ (g 100 g⁻¹) | C:N | TRFe (mg Fe g⁻¹) | AVS (µg S g⁻¹) | CRS (µg S g⁻¹) |
|---|---|---|---|---|---|---|---|---|---|
| **RG1** | | | | | | | | | |
| Depth 1 | 2.5-7.5 | 5.00 (0.09) | 54.6 (14.0) | 37.4 (0.2) | 1.75 (0.00) | 21.3 | 3.63 (0.11) | 2.46 (0.29) | 146.1 (27.6) |
| Depth 2 | 7.5-12.5 | 5.14 (0.14) | 31.5 (3.5) | 38.2 (0.2) | 1.79 (0.01) | 21.3 | 4.03 (0.44) | NA | NA |
| Depth 3 | 17.5-22.5 | 5.27 (0.26) | 79.6 (22.6) | 39.7 (0.3) | 1.80 (0.04) | 22.1 | 4.14 (0.32) | NA | NA |
| Depth 4 | 36-40 | 4.6 (0.07) | 107.1 (22.3) | 43.1 (2.7) | 1.85 (0.03) | 23.3 | 3.04 (0.26) | 2.46 (0.3) | 109.7 (23.5) |
| Depth 5 | 47.5-52.5 | 5.05 (0.18) | 99.8 (26.5) | 31.0 (15.6) | 1.47 (0.64) | 21.1 | 2.50 (0.55) | 3.50 (0.52) | 40.1 (8.9) |
| Depth 6 | 93-98 | 5.32 (0.03) | 41.8 (1.8) | 0.57 (0.3) | 0.01 (0.01) | 47.8 | 0.14 (0.04) | NA | NA |
| **RG2** | | | | | | | | | |
| Depth 1 | 0-25 | 5.00 | 57.6 | 19.8 (3.4) | 1.34 (0.13) | 14.8 | 4.48 (0.11) | NA | NA |
| Depth 2 | 25-50 | 5.14 | 63.2 | 8.9 (3.0) | 0.63 (0.23) | 14.2 | 2.29 (0.25) | 1.71 (0.00) | 33.3 (7.3) |
| **AR1** | | | | | | | | | |
| Depth 1 | 2.5-7.5 | 5.11 (0.09) | 93.4 (6.3) | 35.9 (0.1) | 1.81 (0.02) | 19.9 | 4.57 (0.09) | 3.19 (0.61) | 128.9 (21.5) |
| Depth 2 | 7.5-12.5 | 5.30 (0.08) | 86.8 (6.4) | 34.2 (0.2) | 1.76 (0.02) | 19.4 | 4.66 (0.15) | NA | NA |
| Depth 3 | 17.5-22.5 | 5.16 (0.03) | 80.3 (5.6) | 41.0 (2.2) | 1.93 (0.11) | 21.3 | 4.99 (0.43) | NA | NA |
| Depth 4 | 36-40 | 4.94 (0.29) | 86.4 (4.3) | 41.1 (5.8) | 1.84 (0.05) | 22.4 | 3.23 (0.41) | 2.33 (0.33) | 100.4 (14.8) |
| Depth 5 | 47.5-52.5 | 4.84 (0.14) | 80.8 (13.2) | 5.9 (1.7) | 0.37 (0.13) | 16.3 | 1.19 (0.19) | 2.13 (0.14)[3] | 48.4 (7.4) |
| Depth 6 | 93-98 | 5.49 (0.08) | 65.8 (1.9) | 0.27 (0.1) | 0.00 (0.00) | 68.6 | 0.18 (0.02) | NA | NA |
| **AR2** | | | | | | | | | |
| Depth 1 | 0-25 | 5.11 | 63.5 | 33.38 (1.2) | 1.45 (0.03) | 23.1 | 4.11 (0.07) | 1.65 (0.02) | 44.7 (8.2) |
| Depth 2 | 25-50 | 5.11 | 65.2 | 38.35 (0.2) | 1.46 (0.02) | 26.2 | 3.78 (0.14) | NA | NA |

718

719





**Table 2.** Cumulative emissions of $N_2O$ (kg $N_2O$ ha$^{-1}$). Estimation for each season was performed using the trapezoidal approximation of the integral of the emission curve. Numbers in parentheses indicated 95% confidence intervals, and significant differences, corrected for multiple testing by the single-step method, are indicated by asterisks.

| Spring (99-105 d) | | | *RG* -NF | *RG* -F | *AR* -NF |
|---|---|---|---|---|---|
| *RG* -NF | 2.0 | (1.5-2.5) | | | |
| *RG* -F | 7.3 | (4.9-9.6) | ***§ | | |
| *AR* -NF | 17.1 | (13.9-20.2) | *** | *** | |
| *AR* -F | 15.0 | (12.2-17.8) | *** | *** | NS |
| | | | | | |
| **Autumn (47-69 d)** | | | *RG* | | |
| *RG* | 2.2 | (1.6-2.7) | | | |
| *AR* | 14.8 | (11.6-17.9) | *** | | |

§ ***, $p < 0.001$





**Figure legends**

**Figure 1.** Experimental design at each of the four sites. Three blocks were defined across the site that were centered around piezometers (●). In the longitudinal direction two subplots were defined, one of which received N fertiliser at the rate of the surrounding field. Six collars were installed on either side of piezometers, sampling positions were labelled S1-S6. Sets of 5 diffusion probes for soil gas sampling at 5, 10, 20, 50 and 100 cm depth were installed near collars in block 2 and 3 (sites *RG1* and *AR1*) or block 2 only (sites *RG2* and *AR2*).

**Figure 2.** Intact soil cores from sites *RG1* and *AR1*, collected on 23 April (DOY112) and 2 September (DOY244), were analysed for nitrite-N and total reactive iron (TRFe). The four plots show average concentrations of nitrite-N in *RG1* (A) and *AR1* (C), and TRFe concentrations in *RG1* (B) and *AR1* (D). White symbols: 23 April; Dark symbols: 2 September.

**Figure 3.** Rainfall, air temperature and management at sites *RG1* (left) and *RG2* (right) during spring are shown in the top panel. F: fertilisation, where arrow indicates occurrence outside shown period. The middle section shows $N_2O$ fluxes ($n = 3$) and contour plots of soil profile $N_2O$ concentrations in block 2 and 3 (where available). Similarly, the bottom section shows $N_2O$ fluxes and soil profile $N_2O$ concentrations for unfertilised plots.

**Figure 4.** Rainfall, air temperature and management at sites *AR1* (left) and *AR2* (right) during spring are shown in the top panel. T: tillage; F: fertilisation. The middle section shows $N_2O$ fluxes ($n = 3$) and contour plots of soil profile $N_2O$ concentrations in block 2 and 3 (where available). Similarly, the bottom section shows $N_2O$ fluxes and soil profile $N_2O$ concentrations for unfertilised plots.

**Figure 5.** Rainfall, air temperature and management at sites *RG1* (left) and *RG2* (right) during autumn are shown in the top panel. H: harvest. The middle section shows $N_2O$ fluxes ($n = 3$) and contour plots of soil profile $N_2O$ concentrations in block 2 and 3 (where available). Similarly, the bottom section shows $N_2O$ fluxes and soil profile $N_2O$ concentrations for unfertilised plots.

**Figure 6**. Rainfall, air temperature and management at sites *AR1* (left) and *AR2* (right) during autumn are shown in the top panel. H: harvest. The middle section shows $N_2O$ fluxes ($n = 3$) and contour plots of soil profile $N_2O$ concentrations in block 2 and 3 (where available). Similarly, the bottom section shows $N_2O$ fluxes and soil profile $N_2O$ concentrations for unfertilised plots.

**Figure 7.** Results from graphical models made separately for the four combinations of crops (rotational grass, *RG*, and potato as arable crop, *AR*) and season (spring and autumn). A. *RG*, spring; B. *GR*, autumn; C. *AR*, spring; and D. *AR*, autumn. The vertices ("points") and edges ("lines") indicate significant relationships between explanatory variables and the response variable, i.e., $N_2O$ flux.





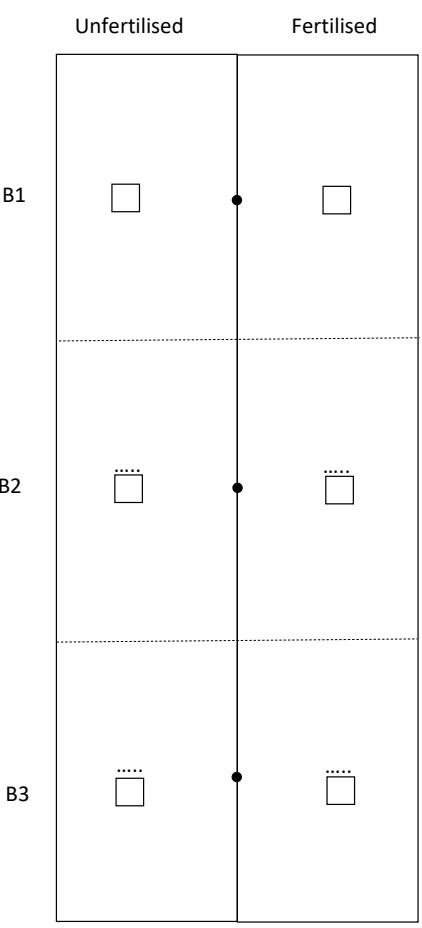

**Figure 1**





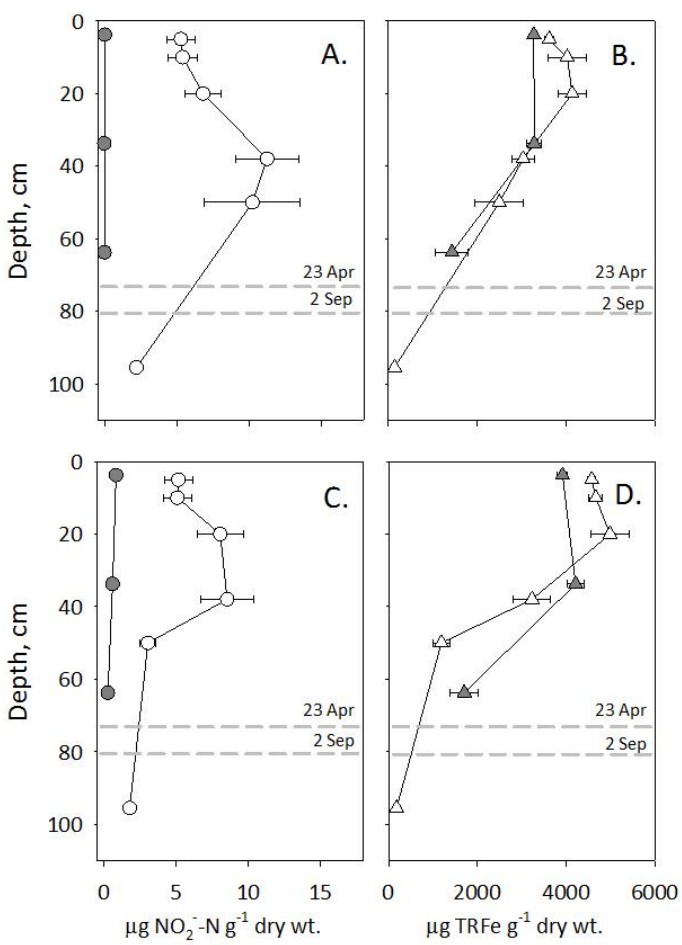

**Figure 2.**



**Figure 3**





**Figure 4**





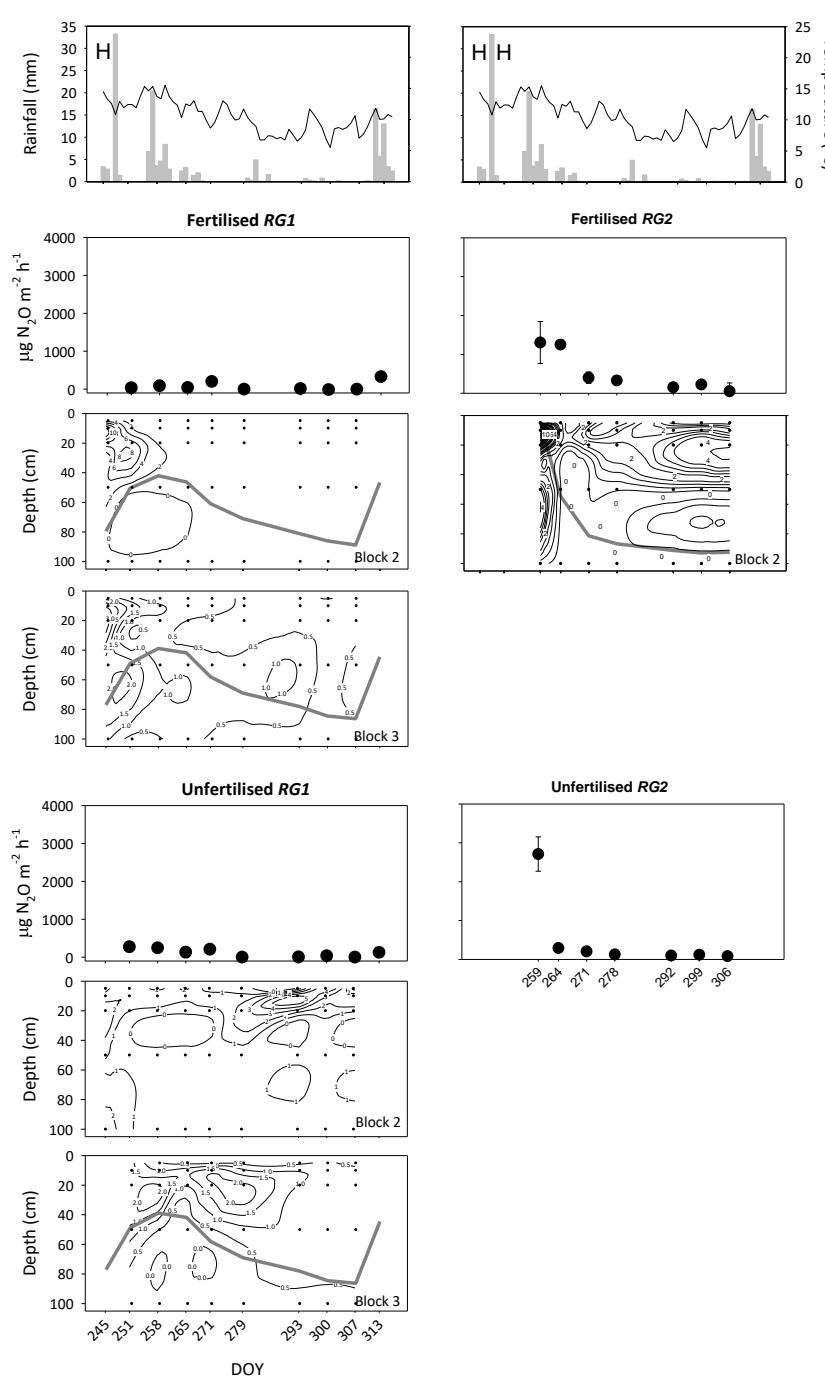

**Figure 5**





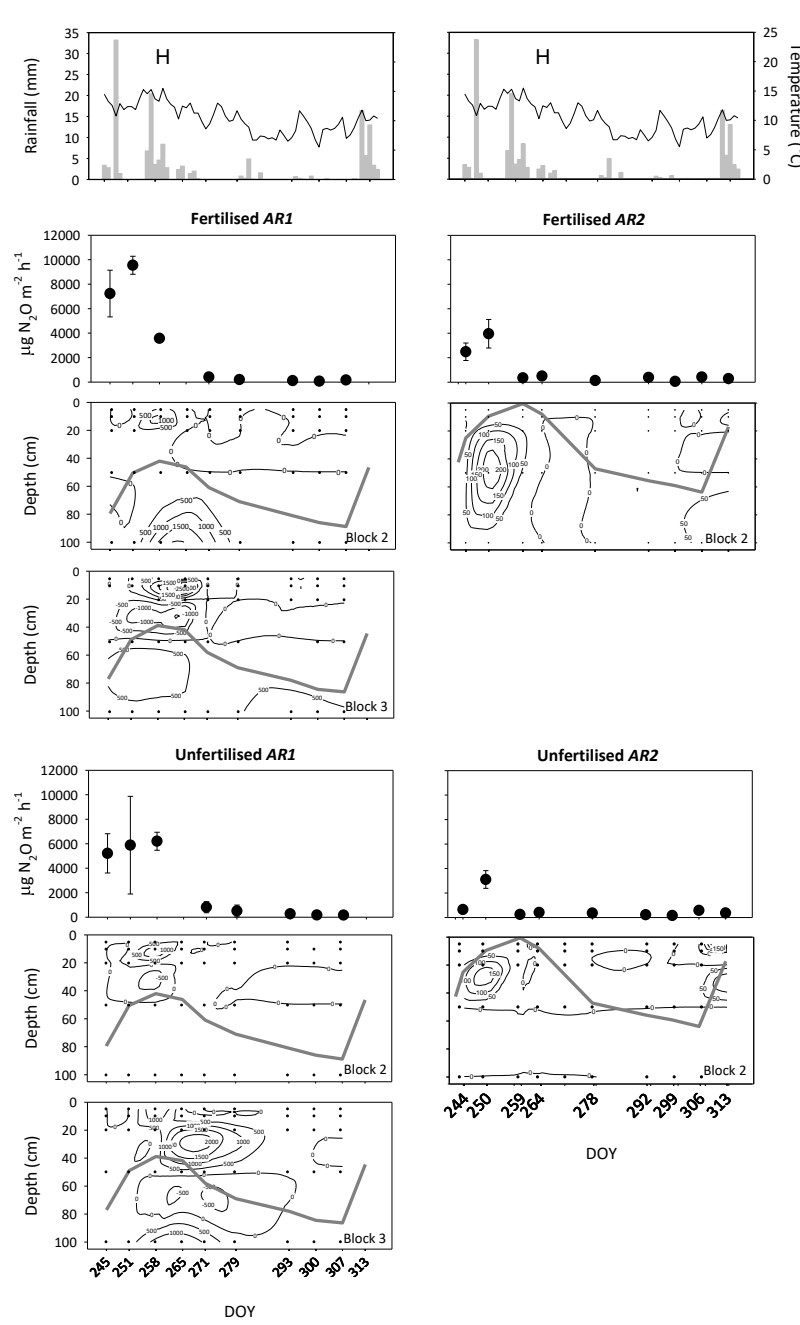

**Figure 6**





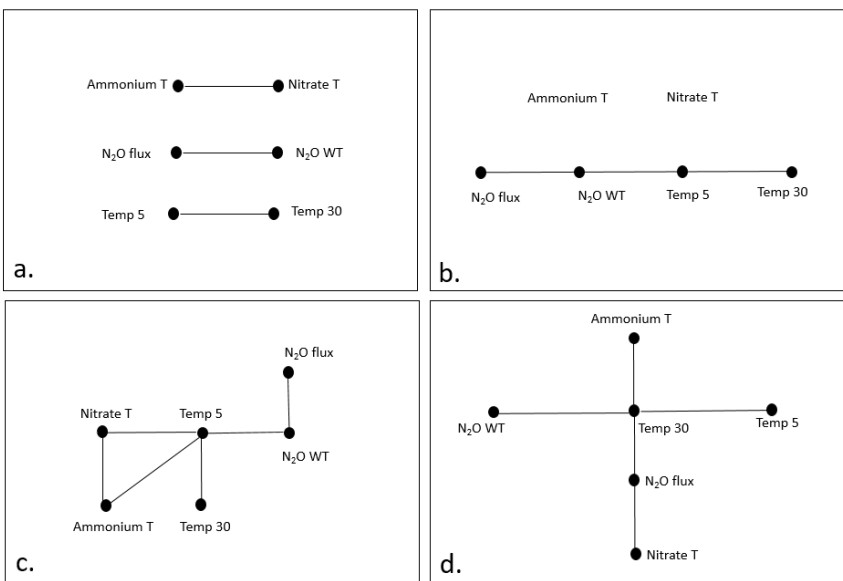

**Figure 7**