# Peer review of "Regulation of N2O emissions from acid organic soil drained for agriculture: Effects of land use and season"

_Biogeosciences, 2018_

## Referee Comment (RC1) · Anonymous Referee #1 · 8 May 2018

Taghizadeh-Toosi and others study N2O emissions from a raised bog in northern Denmark. The study was performed competently but there are many highly speculative statements and the authors seem to continuously want to extend inference beyond what the data allow. Re-writing the paper to emphasize findings versus more speculative concepts that can be addressed in future studies would represent an improvement. Minor comments follow:

14: why is the soil "potentially" acid sulfate? (see also 81).

Lots of speculative statements in the abstract. "probably competition from plants for available N", "iron sulfides were probably the source", "appear to be important controls".

These statements need to be supported or not made at all.

The Introduction is well-written, but would be improved if the "high emissions" on line 68 were described quantitatively, and a hypothesis shouldn't say "possibly" as this is less straightforward to falsify.

103: the soil diffusion probes should be described in more detail rather than merely referring the reader to Petersen, 2014.

Why could precipitation not be measured at the site? If it wasn't measured, that's ok, but I can't think of a technical reason why it wouldn't be able to be measured.

Are fertilization rates typical for the land management practices? And why were measurements only made during morning hours? Is there a diurnal pattern in N2O flux that may be missed as a consequence?

Just spell out weight on line 181.

On 234, was time of day an important factor? Or perhaps better yet temperature? I see these being included later but why not in the GLMM's?

On 252 how are the properties optimal?

315 why is water table depth not shown? This will help the interpretation.

Try to avoid superlatives like 'dramatic' on line 340. Also, qualitative statements like 'low' on 374 and elsewhere are difficult to interpret and need to be removed or made quantitative.

The passage 'spring of 2000-6000 ug N2O m-2 h-1' is confusing on line 371.

'were. . .occurring' on 372 needs to be re-worded.

The argument on 435 about grass and competition with available N needs to be revisited. This is likely the case but you can't definitively say it here, only note that it is consistent with the notion (and the Schothorst 1977 reference seems to me to be a bit

of a stretch to use in its justification). Perhaps the competition is an important course of future study and that results point toward it. Likewise the statement on 459 is highly speculative about the minearalization of N in potato crop residues. Grasslands also have lots of residues from deceased grass.

The major finding of the graphical model on 468 is that N2O diffuses if concentration at depth is the most important factor. This is not a novel finding.

'Rainfall most likely triggered' could be tested using the dataset.

The passage on 519 is entirely speculative. The (speculative) section 4.5 is well-written, but it seems like the authors want to push their findings past the ability of the data to make them. This needs to be reconciled. AOA activity is a hypothesis for future studies, not to argue for given other studies including those from deep-sea water columns!

Figure 1 is not informative. Please make a real map.

Many if not all figures would benefit from readable font sizes.

---

## Referee Comment (RC2) · Anonymous Referee #2 · 18 May 2018

General Comments:

This is a manuscript by Taghizadeh-Toosi and colleagues looking at N2O emissions, and driving factors, from drained histosols. This paper has some potential and interesting concepts within it. It could be of interest to readers of Biogeosciences. However, it has some major flaws. I'm not quite sure what the authors objectives of the paper are. They "searched for relationships" in N2O emissions, but why did they use two different fields? They hypothesized that N2O would be produced in the capillary fringe, but then they didn't show any data to confirm or deny this hypothesis. Where were the measurements in the capillary fringe and how did they confirm this? Not to mention

there is a focus on season, but they only measured N2O for two seasons and just a couple measurements in each season. This is not enough to claim a "season effect". The manuscript itself is loosely held together, figures are difficult to interpret, and the writing is poor (See specific comments).

Specific Comments:

L10. The Abstract seems too long. Consider shortening and making more clear and concise.

L18. Change 'recorded' to 'measured'

L24. 'In connection' is not the appropriate term here.

L43. Delete 'depth'

L44-45. What about CH4? Seems like the shift in CH4 production/consumption could offset some of the new losses when drying a peat soil?

L51. What do you mean by 'site conditions'? This is very vague.

L66. Change 'soil conditions' to 'soils. Replace 'with' with 'under'. Delete 'in the experimental year'

L70. Change 'pursued the hypotheses' to 'hypothesized'

L83. What does 'after field trips and meetings with farmers' have to do with how the field sites were distributed? Could you be more specific?

L93. Figure one is not very informative. Please change. Show more details or a key.

L116. Add 'e.g.' after 1st parentheses. Move parenthetical statement after 'farmers'

L255. Delete 'monitoring period'

L256. Delete 'monitoring period'

L267. Add 's' after 'soil', and change 'was' to 'were'

[Figure]

L283-284. Delete sentence.

L305. Delete sentence.

L326-328. Delete sentence. It is not necessary to tell where data is shown. Just describe the data and then put the Figure number in parentheses.

L337-338. Why is this sentence out by itself? It should be in a paragraph

L361-363. Delete sentence. Same as previous comment.

---

## Author Comment (AC1) · 4 Jun 2018

Anonymous Referee #1 Taghizadeh-Toosi and others study N2O emissions from a raised bog in northern Denmark. The study was performed competently but there are many highly speculative statements and the authors seem to continuously want to extend inference beyond what the data allow. Rewriting the paper to emphasize findings versus more speculative concepts that can be addressed in future studies would represent an improvement.

Response: Thank you for this comment, and for reviewing our paper. The present study was planned to examine in more detail observations from an earlier study (Petersen et al., 2012) – more specifically, apparent interactions between N availability and WT dynamics with respect to N2O emissions – and therefore it was by nature exploratory. In general, there are still large uncertainties regarding the mechanisms and pathways of N2O emissions from drained peat soils. Here we have offered an interpretation consistent with the above- and belowground dynamics of N2O observed, as well as soil characteristics. A graphical model analysis of potential drivers provided a quantitative analysis of relationships. We have indeed referred extensively to other literature to support our data interpretation, and to relate these new observations to existing knowledge, and this may be why some statements have come across as speculative. However, we will critically review and revise the text to ensure a better balance between own findings and discussion of the wider context.

Reference: Petersen, S.O., Hoffmann, C. C., Schäfer, C.-M., Blicher-Mathiesen, G., Elsgaard, L., Kristensen, K., Larsen, S. E., Torp, S. B., and Greve, M. H., 2012. Annual emissions of CH4 and N2O, and ecosystem respiration, from eight organic soils in Western Denmark managed by agriculture. Biogeosciences, 9, 403-422, doi: https://doi.org/10.5194/bg-9-403-2012.

Minor comments follow: 14: why is the soil "potentially" acid sulfate? (see also 81).

Response: We refer to the terminology used by Madsen et al. (1988) cited in the paper. The distribution of iron sulfides in soil is heterogeneous, and the classification is based on soil sampling and analysis, and hence the classification is based on frequencies of occurrence. In the classification from the 1980s (www2.mst.dk/Udgiv/publikationer/1984/87-88613-03-8/pdf/87-88613-03-8.pdf; in Danish), the area of the present study was characterised as a potentially acid sulfate soil.

Reference: Madsen, H.B. and Jensen, N.H., 1988. Potentially acid sulfate soils in relation to landforms and geology. Catena, 15, 137-145, doi: https://doi.org/10.1016/0341-8162(88)90025-2.

Lots of speculative statements in the abstract. "probably competition from plants for available N", "iron sulfides were probably the source", "appear to be important controls". These statements need to be supported or not made at all.

Response: In fact the statements referred to above were supported, either by findings of this study (iron oxides/hydroxides as the main sources of reactive iron; controlling WT depth and N supply could mediate against N2O emissions), or by literature cited in the discussion (a previous study had provided evidence for extensive N uptake by a grass sward). In view of the fact that we refer to field observations rather than results from controlled experiments, we believe it is good scientific practice not to make strong statements about causal relationships. We will, however, remove the statement referring to literature results from the abstract, to clarify what was observed in this study.

The Introduction is well-written, but would be improved if the "high emissions" on line 68 were described quantitatively, and a hypothesis shouldn't say "possibly" as this is less straightforward to falsify.

Response: We will change text to read "...where N2O emissions in excess of 2 mg m-2 h-1 were consistently observed." We will further revise the hypothesis in order to avoid the term "possibly".

103: the soil diffusion probes should be described in more detail rather than merely referring the reader to Petersen, 2014.

Response: The original text did include a description of the soil gas diffusion probe design, i.e., "The stainless steel probes were constructed as described by Petersen (2014): with a 10 cm3 cell connected to the surrounding soil via a 3 mm diameter opening at the sampling depth that was covered by a silicone membrane, and connected to the soil surface via two 18G steel tubes with Luer Lock fittings (Petersen, 2014)." (L.105-108) Also, the sampling procedure is described in some detail (L160-168). Any additional detail would only provide technical details such as suppliers of components, as in the method paper (Petersen, 2014). We feel this will not add to the

clarity of the text and would prefer not to elaborate further. We will, however, include photo documentation of probes and sampling procedure as online 'Supplementary Information'.

Reference: Petersen, S.O., 2014. Diffusion probe for gas sampling in undisturbed soil. Eur. J. Soil Sci., 65, 663-671, doi:10.1111/ejss.12170.

Why could precipitation not be measured at the site? If it wasn't measured, that's ok, but I can't think of a technical reason why it wouldn't be able to be measured.

Response: We apologize for the unfortunate wording. Equipment used in the previous monitoring program (Petersen et al., 2012) was no longer available, and resources for the present project did not allow for a new investment. Instead we used measurements from the nearby meteorological station. The sentence will be revised to remove ambiguity.

Are fertilization rates typical for the land management practices? And why were measurements only made during morning hours? Is there a diurnal pattern in N2O flux that may be missed as a consequence?

Response: Yes, fertilization and other management followed the field operations of the individual fields (cf. section "2.3. Management") except where stated. Mid- to late morning periods were selected for samplings, since previous studies have indicated that the N2O flux at this time of day is often close to the daily average flux (Laville et al., 2011). Soil temperature is an important driver for N2O emissions and may indicate the potential error in the present study. We have therefore calculated deviations between soil temperature at 5 cm depth at sampling and the 24-hour mean for all sampling days at each of the four locations. The mean deviations ranged from 0.2 to 0.85 degree C, and the overall largest deviations were -2.0 and +2.1 degree C. Surface emissions lag behind the time of N2O production at depth in the soil (Clough et al. 1999), but temperature variations would also be dampened. We therefore believe, in agreement with current recommendations (de Klein and Harvey, 2015), that mid-morning N2O flux

measurements were representative for daily mean fluxes.

References: Clough, TJ, Jarvis, SC, Dixon, ER, Stevens, RJ, Laughlin, RJ & Hatch, DJ, 1999, 'Carbon induced subsoil denitrification of 15N-labelled nitrate in 1-m deep soil columns', Soil Biology and Biochemistry. 31, 31-41.

De Klein, C.A.M. & Harvey, M. (ed.), 2015. Nitrous Oxide Chamber Methodology Guidelines. Ministry of Primary Industries, Wellington, New Zealand. 146 pp.

Laville, P, Lehuger, S, Loubet, B, Chaumartin, F, & Cellier, P, 2011, 'Effect of management, climate and soil conditions on N2O and NO emissions from an arable crop rotation using high temporal resolution measurements', Agricultural and Forest Meteorology. 151, 228-240.

Just spell out weight on line 181.

Response: Thank you. "wt." will be changed to "weight".

On 234, was time of day an important factor? Or perhaps better yet temperature? I see these being included later but why not in the GLMM's?

Response: We refer to the previous answer with respect to the potential effect of 'time of day' (i.e., temperature). Temperature was initially included in GLMM analysis, but removed because it was not a significant factor.

On 252 how are the properties optimal?

Response: The optimal property referred in the text concerns the use of the BIC (Bayesian Information Criterium), i.e., the log-likelihood minus the number of parameters multiplied by the logarithm of the number of observations. Haughton (1988) proved that, when using this specific penalization of the log-likelihood function, the probability of choosing a correct model (i.e., a correct graphical representation) tends to one as the number of observations increases. Note that this property is not shared by other penalization procedures. This point will be clarified in the text.

315 why is water table depth not shown? This will help the interpretation.

Response: Since measurements of continuous water table depth were only available for potato crops in the autumn part of the study, we focused on the weekly recorded WT depths available at all sites in both seasons.

Try to avoid superlatives like 'dramatic' on line 340. Also, qualitative statements like 'low' on 374 and elsewhere are difficult to interpret and need to be removed or made quantitative.

Response: We will review the text and refer to quantitative information where possible. The results are also presented in Figures and Tables, and therefore, in order to ensure clarity of the text, in some cases a proper qualitative term may be appropriate.

The passage 'spring of 2000-6000 ug N2O m-2 h-1' is confusing on line 371.

Response: We will reword this sentence to: "Fluxes during early spring reached 2000-6000 ug N2O m-2 h-1 and were higher than in late spring where, as for site RG1, no effect of N fertilization was observed."

'were. . .occurring' on 372 needs to be re-worded.

Response: We will reword this sentence to: "The highest observations occurred, independent of fertilization, in June when a WT rise to 30 cm depth was observed."

The argument on 435 about grass and competition with available N needs to be revisited. This is likely the case but you can't definitively say it here, only note that it is consistent with the notion (and the Schothorst 1977 reference seems to me to be a bit of a stretch to use in its justification). Perhaps the competition is an important course of future study and that results point toward it. Likewise the statement on 459 is highly speculative about the minearalization of N in potato crop residues. Grasslands also have lots of residues from deceased grass.

Response: The conclusion that the grass sward took up N at significant soil depths

was not our interpretation, but the conclusion of Schothorst (1977), which will be made clear. We would like to include the specific information from this widely cited study in order to highlight the importance of vegetation for N availability in peat soil. More specific references to the supporting data on soil mineral N in Table S1 vs. S2 will be made. They show consistently low soil mineral N concentrations at 25-50 cm depth under grassland (site RG1) compared to the neighbouring site used for a potato crop (site AR1). With respect to the statement that potato crop residues were a main source of soil mineral N, we will again refer more specifically to the temporal dynamics of soil mineral N as revealed in Table S1 and S2 for sites RG1 and AR1, respectively. We will include quantitative information about the accumulation of mineral N in early autumn at the two adjoining sites.

The major finding of the graphical model on 468 is that $N_2O$ diffuses if concentration at depth is the most important factor. This is not a novel finding.

Response: The main result here was not that $N_2O$ accumulation at depth in the soil was a main driver for $N_2O$ emissions, but that this was not always the case. The graphical model analysis found that, in the case of the potato crop in autumn, nitrate accumulation (and temperature) were more important in predicting $N_2O$ emissions. We take this as evidence that $N_2O$ emissions at grassland sites, and across the two potato sites during spring, were mainly controlled by the mineralization of N from decomposing peat. Following potato harvest, nitrate accumulated and, apparently, became the limiting factor after rainfall. We believe this interaction between season and land use, and the evidence for the different controls of $N_2O$ emissions, is new and interesting.

'Rainfall most likely triggered' could be tested using the dataset.

Response: The inclusion of rainfall as an independent factor is complicated by the lack of information about the temporal dynamics of WT depth after rainfall, and the change in water-filled porosity above the capillary fringe. Rainfall will induce an increase in WT depth, and in our analysis this was indirectly represented by determining soil $N_2O$ con-

centrations closest to, but above the WT depth. The inclusion of rainfall (or WT depth) as an independent variable would weaken the relationship between N2O accumulation in the capillary fringe and N2O emissions.

The passage on 519 is entirely speculative. The (speculative) section 4.5 is well written, but it seems like the authors want to push their findings past the ability of the data to make them. This needs to be reconciled. AOA activity is a hypothesis for future studies, not to argue for given other studies including those from deep-sea water columns!

Response: We believe our results support the hypothesis that N2O was produced both above and below the WT depth, and we wanted to discuss possible pathways based on existing knowledge. On the other hand, we acknowledge that we may have taken the discussion a step too far by including examples beyond soil environments. The paragraph starting with L. 519 will be omitted, and Section 4.5 as a whole will be carefully scrutinized to avoid speculation without a firm link to findings and relevant literature.

Figure 1 is not informative. Please make a real map.

Response: We will extend Figure 1 to include a map showing the geographic distribution of sampling locations. However, we would like to also include the simple outline of the layout of each site in order to show the relative positions of gas chambers, gas diffusion probes, and piezometers among the three blocks, and the division among fertilised and non-fertilised subplots.

Many if not all figures would benefit from readable font sizes.

Response: We will inspect all Figures and increase font sizes where possible, i.e., subplots in composite Figures must still be clearly separated. The contour plots represent a particular challenge, since here the resolution and font size for concentrations must be balanced. We have indeed spent time experimenting with this and find that the size already used is the best possible compromise.

Anonymous Referee #2 General Comments: This is a manuscript by Taghizadeh-Toosi and colleagues looking at N2O emissions, and driving factors, from drained histosols. This paper has some potential and interesting concepts within it. It could be of interest to readers of Biogeosciences. However, it has some major flaws. I'm not quite sure what the authors objectives of the paper are. They "searched for relationships" in N2O emissions, but why did they use two different fields? They hypothesized that N2O would be produced in the capillary fringe, but then they didn't show any data to confirm or deny this hypothesis. Where were the measurements in the capillary fringe and how did they confirm this? Not to mention there is a focus on season, but they only measured N2O for two seasons and just a couple measurements in each season. This is not enough to claim a "season effect". The manuscript itself is loosely held together, figures are difficult to interpret, and the writing is poor (See specific comments).

Response: Thank you for this comment, and for the evaluation of our paper. We would like to start by reiterating that this study was developed from observations made in a 14-month long monitoring study of N2O, CH4 and CO2 fluxes of eight organic soils used for agriculture (Petersen et al., 2012); this study included the sites RG1 and AR1, but new sites RG2 and AR2 were included in this new study for verification of land use effects. The seasonal pattern in N2O emissions, and effects of land use, had therefore already been documented and were not objectives of this study. Based on the previous results we hypothesized that WT dynamics and soil mineral N status were important drivers of N2O emissions, and the present study focused on these periods with high N2O emissions. Compared to the previous study, we changed the sampling frequency from three weeks to one week, and we included new sites to conrfim the trends observed previously at sites RG1 and AR1. This was an exploratory study, and we analysed relationships between N2O emissions and soil characteristics in search of important drivers. Two land uses were included, because the limited observations made in the previous study (Petersen et al., 2012; Table 4) had indicated that soil mineral N availability would be different in these periods. With this experimental design

we hoped to be able to separate the effect of WT depth from the effect of mineral N availability. The comment regarding evidence for N2O production in the capillary fringe must be based on a misunderstanding, since we measured equivalent soil gas phase concentrations of N2O at 5, 10, 20, 50 and 100 cm depth (see contour plots in Figures 3 to 6). For the graphical model analysis, we filtered out N2O concentrations immediately above the (fluctuating) WT depth, and indeed the analysis found that this variable was strongly related to N2O emissions during spring with both land uses, as well as also in the autumn for site RG1.

Specific Comments: L10. The Abstract seems too long. Consider shortening and making more clear and concise.

Response: We will revise the abstract to improve conciseness and focus on findings from the study.

L18. Change 'recorded' to 'measured'

Response: We would like to keep the broader term "recorded" as a verb here, since the information gathered consisted of a variety of measurements, including weather station data.

L24. 'In connection' is not the appropriate term here.

Response: We will change to "after rainfall".

L43. Delete 'depth'

Response: Will be done.

L44-45. What about CH4? Seems like the shift in CH4 production/consumption could offset some of the new losses when drying a peat soil?

Response: The previous study (Petersen et al., 2012) showed that CH4 fluxes from arable soil and rotational grass were low throughout the year, and CH4 was therefore not considered in the present study.

L51. What do you mean by 'site conditions'? This is very vague.

Response: In fact the next sentence defines site conditions: "Site conditions are defined by land use, management, inherent soil properties and climate (Mander et al., 2010; Leppelt et al., 2014)." (L51-52) Further, L. 52-58 provides examples of relationships between individual soil characteristics and N2O emissions.

L66. Change 'soil conditions' to 'soils. Replace 'with' with 'under'. Delete 'in the experimental year'

Response: We respectfully argue to keep the exact wording here. 'Acid soil' is a generalising term, but the four sites differed in several respects besides pH. The potato crop was established and harvested in the middle of the spring and autumn measurement periods, respectively, and therefore 'under a potato crop' would not be accurate.

L70. Change 'pursued the hypotheses' to 'hypothesized'

Response: Will be done.

L83. What does 'after field trips and meetings with farmers' have to do with how the field sites were distributed? Could you be more specific?

Response: Poor wording, will be changed to: "Following field trips and meetings with farmers, four field sites were found that were distributed along an east-west transect."

L93. Figure one is not very informative. Please change. Show more details or a key.

Response: Key to elements will be included in the Figure. As stated above, a map of the area showing distribution of sites will also be included.

L116. Add 'e.g.' after 1st parentheses. Move parenthetical statement after 'farmers'

Response: Will be done.

L255. Delete 'monitoring period'

Response: We prefer to keep this term, since 'spring' is not a well-defined period, and

we would like to emphasize conditions specific to our monitoring periods.

L256. Delete 'monitoring period'

Response: Please see the previous response regarding spring.

L267. Add 's' after 'soil', and change 'was' to 'were'

Response: Will be done.

L283-284. Delete sentence.

Response: We understand the reviewer's intent to improve conciseness by avoiding sentences which guide the reader to specific results. We will revise the text in order to integrate references to Figures and Tables in sentences describing the results.

L305. Delete sentence.

Response: Please see comment to L. 283-284.

L326-328. Delete sentence. It is not necessary to tell where data is shown. Just describe the data and then put the Figure number in parentheses.

Response: Please see comment to L. 283-284.

L337-338. Why is this sentence out by itself? It should be in a paragraph

Response: This sentence referred to results from the autumn measurement period. Although the paragraph consists of only one statement, we would like to avoid merging spring and autumn results into one paragraph, since the results showed qualitative differences.

L361-363. Delete sentence. Same as previous comment.

Response: Done.

Please also note the supplement to this comment:

https://www.biogeosciences-discuss.net/bg-2018-9/bg-2018-9-AC1-supplement.pdf

---

## Author Comment (AC3) · 4 Jun 2018

There was a typing error on the title, the word "response". I apologize.

---

## Author Comment (AC4) · 4 Jun 2018

There was a typing error on the title, the word "response". I apologize.